# Gentian Violet Inhibits Cell Proliferation through Induction of Apoptosis in Ovarian Cancer Cells

**DOI:** 10.3390/biomedicines11061657

**Published:** 2023-06-07

**Authors:** Min Sung Choi, Ji Hyeon Kim, Chae Yeon Lee, Yul Min Lee, Sukmook Lee, Ha Kyun Chang, Hyun Jung Kim, Kyun Heo

**Affiliations:** 1Department of Biopharmaceutical Chemistry, Kookmin University, Seoul 02707, Republic of Korea; k1choi4@kookmin.ac.kr (M.S.C.); minyzoa@kookmin.ac.kr (Y.M.L.); lees2018@kookmin.ac.kr (S.L.); 2Biopharmaceutical Chemistry Major, School of Applied Chemistry, Kookmin University, Seoul 02707, Republic of Korea; kjh0808@kookmin.ac.kr (J.H.K.); panda3364@kookmin.ac.kr (C.Y.L.); 3Antibody Research Institute, Kookmin University, Seoul 02707, Republic of Korea; 4Department of Obstetrics and Gynecology, Korea University Ansan Hospital, Korea University College of Medicine, Ansan 15855, Republic of Korea; coolblue23@naver.com

**Keywords:** gentian violet, ovarian cancer, apoptosis

## Abstract

Gentian violet (GV) is known to have antibacterial and antifungal effects, but recent studies have demonstrated its inhibitory effects on the growth of several types of cancer cells. Here, we investigated the anticancer efficacy of GV in ovarian cancer cells. GV significantly reduced the proliferation of OVCAR8, SKOV3, and A2780 cells. Results of transferase dUTP nick and labeling (TUNEL) assay and Western blot assay indicated that the inhibitory effect of GV on ovarian cancer cells was due to the induction of apoptosis. Moreover, GV significantly increased reactive oxygen species (ROS) and upregulated the expression of p53, PUMA, BAX, and p21, critical components for apoptosis induction, in ovarian cancer cells. Our results suggest that GV is a novel antiproliferative agent and is worthy of exploration as a potential therapeutic agent for ovarian cancer.

## 1. Introduction

Ovarian cancer (OC) accounts for 3.4% of all cancers in women and has the eighth highest incidence rate among all cancers globally [1]. OC is the fifth leading cause of cancer-related death in women after lung, breast, colon, and pancreatic cancers; in 2022, there were an estimated at 19,880 newly diagnosed cases of OC and 12,810 deaths due to OC in the United States [2]. Owing to the lack of specific symptoms in the early stages and the paucity of effective diagnostic methods and specific markers, most patients with OC are diagnosed at an advanced stage of the disease. More than 60% of newly diagnosed OC patients have stage III or IV disease with peritoneal, retroperitoneal, and inguinal lymph node metastasis. Among gynecologic malignancies, OC is considered one of the most dangerous, showing a high fatality rate and relapse due to chemoresistance [3]. The 5-year survival rates in patients with stage III and IV OC (41% and 20%, respectively) are considerably lower than in patients with stage I and II OC (81% and 71%, respectively) [4,5]. Development of novel markers may help improve the survival rate of these patients.

Optimal debulking surgery and chemotherapy are currently the main treatment modalities for patients with advanced-stage OC. For decades, the combination of carboplatin and paclitaxel has been the standard chemotherapy for OC. In 2014, bevacizumab, marketed under the trade name Avastin^®^, was approved by the FDA for the treatment of advanced-stage OC, and is used in combination with carboplatin and paclitaxel [6]. Bevacizumab, a humanized monoclonal antibody of vascular endothelial growth factor (VEGF) A, binds to, and neutralizes VEGF A, followed by the inactivation of the VEGF receptor. Inactivation of VEGF receptors leads to the formation of abnormal blood vessels and limits blood recruitment, eventually leading to cell death. Despite these effects, bevacizumab increases the overall survival of patients only by 20–30% [7,8,9]. Poly ADP-ribose polymerase (PARP) inhibitors, approved by the FDA in 2020, are currently being used to treat OC. PARP is a protein involved in a number of cellular functions, including cell death induction, and DNA single-strand break repair. PARP inhibitors inhibit PARP from repairing DNA single-strand breaks, whereby unrepaired DNA single-strand breaks form DNA double-strand breaks. In normal cells, DNA double-strand breaks are repaired by homologous recombination (HR); however, breast cancer susceptibility (*BRCA*) gene-mutated cancer cells are unable to repair DNA double-strand breaks through the HR repair system. Consequently, PARP inhibitors induce apoptosis through DNA damage accumulation in patients with *BRCA*-mutant OC. Although the use of PARP inhibitors was found to reduce the risk of progression or death in patients with *BRCA* mutations by 70%, PARP inhibitors are effective only in HR-deficient patients, which account for 50% of OC patients; the remaining 50% of OC patients are resistant to it. Therefore, there is a need to develop new therapies for OC [10,11,12].

Gentian violet (GV), also known as crystal violet, was used as an antibacterial and antifungal agent until the discovery of penicillin in the 1940s. In addition, GV is also used for bacterial cell staining to distinguish between Gram-positive and -negative bacteria. Currently, GV is being studied as a treatment for skin diseases and as an anti-angiogenic and antitumor drug [13,14]. Studies have demonstrated the anticancer effects of GV in several cancer cells; however, its anticancer effects have not yet been demonstrated in OC cells [15,16,17,18,19]. In this study, we observed that GV inhibits the proliferation of OC cells through the induction of apoptosis by using terminal deoxynucleotidyl transferase dUTP nick and labeling (TUNEL) assay and Western blot analysis of apoptotic marker proteins. In addition, GV increased reactive oxygen species (ROS) and p53, PUMA, BAX, and p21 proteins in OC cells. Our results suggest that GV has an antiproliferative effect through apoptosis and has potential as an OC treatment through drug repositioning.

## 2. Materials and Methods

### 2.1. Cell Culture

SKOV3 and A2780 human epithelial ovarian adenocarcinoma cell lines were purchased from KBCC (Korea Biotechnology Commercialization Center, Incheon, Republic of Korea), and the OVCAR8 cell line was obtained from the American Type Culture Collection (Rockville, MD, USA). All cell lines were maintained in Roswell Park Memorial Institute medium (Thermo Fisher Scientific; Waltham, MA, USA) containing 10% heat-inactivated fetal bovine serum (Gibco, Carlsbad, CA, USA), 100 U/mL penicillin, and 100 μg/mL streptomycin (Gibco, Carlsbad, CA, USA). All cells were cultured at 37 °C in a humidified incubator with 5% CO_2_.

### 2.2. Cell Proliferation Assay

GV was purchased from TargetMol (Boston, MA, USA). Cell proliferation was measured using a WST-1 Cell Proliferation Assay Kit (Takara, Kyoto, Japan) according to the manufacturer’s instructions. In brief, 5 × 10^3^ OVCAR8 or A2780 cells or 2 × 10^3^ SKOV3 cells in 200 μL culture medium were seeded onto 96-well tissue culture plates. After 24 h, the cells were treated with 0, 0.3, 1, or 3 μm of GV. After incubation for 24, 48, or 72 h, 20 μL of WST-1 reagent was added to each well and the cells were incubated for an additional 1 h at 37 °C. The proportion of surviving cells was determined by measuring the absorbance of the formazan product at 450 nm using a microplate reader (Synergy H1, BioTek, Winooski, VT, USA). IC_50_ values were calculated by fitting dose–response curves to a four-parameter, variable slope sigmoid dose–response model in GraphPad Prism 5.0 software (GraphPad software Inc., San Diego, CA, USA).

### 2.3. TUNEL Assay

TUNEL assay was performed using the in situ cell death detection kit, fluorescein, according to the manufacturer’s instructions (Roche, Meylan, France) with minor modifications. Briefly, 4 × 10^4^ OVCAR8 and SKOV3 cells were seeded onto 8-well chamber slides (Thermo Fisher Scientific; Waltham, MA, USA) coated with 1 μg/mL of poly-L-lysine (Sigma-Aldrich, St. Louis, MO, USA). After 24 h, the cells were treated with 0, 0.3, 1, or 3 μm of GV. After incubation for 24 h, the cells were fixed in PBS (Gibco, Carlsbad, CA, USA) containing 4% paraformaldehyde (Tech and innovation, Chunchen, Republic of Korea) for 10 min and permeabilized by 0.1% Triton X-100 (Sigma-Aldrich, St. Louis, MO, USA) in PBS for 3 min at room temperature (RT). Cells were then resuspended in 100 μL TUNEL-reaction mixture for 1 h at 37 °C and stained with Hoechst 33342 (Invitrogen, Carlsbad, CA, USA) for 3 min. After washing the slides twice with PBS, the slides were treated with a mounting medium (Vector Laboratories, Burlingame, CA, USA) and examined using a confocal laser scanning microscope (Leica, Wetzlar, Germany). TUNEL intensity of a given cell was then quantified using Image J software (version 1.54d) (NIH, Bethesda, MD, USA) and TUNEL fluorescence intensity was calculated by dividing the mean intensity of apoptotic cells by their control.

### 2.4. Western Blot Analysis

A total of 2 × 10^5^ OVCAR8 and SKOV3 cells in 6-well plates were lysed with RIPA buffer (25 mM Tris-HCl pH 7.6, 150 mM NaCl, 1% NP-40, 1% sodium deoxycholate, 0.1% SDS; Thermo Fisher Scientific) and the total protein amount was determined using the BCA Protein Assay Kit (Thermo Fisher Scientific). A total of 10 μg OVCAR8 and SKOV3 protein samples in cell extracts were separated by 12% SDS-PAGE and transferred to nitrocellulose membranes (Amersham, Little Chalfont, UK). Membranes were blocked with 5% bovine serum albumin (Merck, Burlington, MA, USA) in Tris-buffered saline plus 0.1% (*v*/*v*) Tween-20 (TBST) (Bio-sesang, Yongin, Republic of Korea) for 1 h at RT and then incubated overnight with primary antibodies against PARP (1:1000; #9542), cleaved PARP (1:1000; #5625), caspase-3 (1:1000; #14220), cleaved caspase-3 (1:200; #9664), p53 (1:1000; #2527), PUMA (1:1000; #12450), BAX (1:1000; #5023), p21 (1:1000, #2947) (Cell Signaling Technology, Danvers, MA, USA), and β-actin (1:1000, #sc-47778) (Santa Cruz, Heidelberg, Germany) at 4 °C. After washing six times with TBST, membranes were incubated with horseradish peroxidase-conjugated secondary antibody (Sigma-Aldrich, St. Louis, MO, USA) for 1 h at RT. Membranes were then washed six times with TBST, and chemiluminescence was detected using enhanced chemiluminescence (ECL; Thermo Fisher Scientific) followed by imaging using Amersham ImageQuant 800 (Cytiva, Logan, UT, USA). The bands on the Western blots were quantified by densitometry analysis using the Image J software (https://imagej.nih.gov/ij, accessed on 30 May 2023), National Institutes of Health, Bethesda, MD, USA).

### 2.5. Measurement of ROS Generation

A total of 2 × 10^5^ OVCAR8 and SKOV3 cells were seeded onto 6-well plates. After 24 h, cells were treated with increasing doses of GV (0.18, 0.37, 0.75, 1.5, and 3 μm). After incubation for 24 h, cells were washed with PBS and incubated with 10 μm H_2_DCFDA (Invitrogen, Carlsbad, CA, USA) for 30 min at 37 °C. After washing three times with PBS, cells were lysed with RIPA buffer, and ROS production was determined using a microplate reader (Synergy H1, BioTek, Winooski, VT, USA) at excitation and emission wavelengths of 485 and 530 nm, respectively.

### 2.6. Statistical Analyses

All statistical analyses were performed using the GraphPad Prism software (GraphPad Software Inc., San Diego, CA, USA). Statistical differences between groups were assessed using Student’s *t*-test. The *p*-values < 0.05 were considered statistically significant.

## 3. Results

### 3.1. GV Inhibits Proliferation of OC Cells

In order to determine whether GV inhibits proliferation of OC cells, three OC cell lines (OVCAR8, SKOV3, and A2780) were treated with various concentrations of GV for 72 h and cell growth was measured using the WST-1 cell proliferation assay. GV caused a significant dose-dependent decrease in the growth of all three cell lines (IC_50_ values; 0.665, 0.5867, and 1.0367 μm, respectively) (Figure 1A–F). In particular, treatment with 3 μm GV caused complete inhibition of growth in all three OC cell lines (Figure 1D–F). The results suggested that GV inhibited OC cell proliferation in a dose- and time-dependent manner.

### 3.2. GV Induces Apoptosis in OC Cells

To investigate whether the cell growth inhibition effect by GV in OC cells is due to apoptosis, we first performed TUNEL assay to determine the presence of DNA fragmentation. In both OVCAR8 and SKOV3 cells, TUNEL-positive populations were increased by GV treatment in contrast to the negative control (Figure 2A,B).

Next, Western blot analysis revealed that GV induced the cleavage of caspase-3 and PARP (Figure 3A,B). These data indicated that GV induced the cleavage of caspase-3 and PARP, resulting in apoptosis in OC cells.

### 3.3. GV Increases ROS Levels in OC Cells

Several studies have found that GV increases cellular ROS levels in cancer cells, which is very important for apoptosis [15,17,18]. We therefore investigated the effect of GV on the cellular ROS levels in OC cells. The results indicated that GV significantly increased the ROS levels in a dose-dependent fashion in both OVCAR8 and SKOV3 cells (Figure 4A,B).

In order to investigate the downstream molecules of ROS, we examined the changes in the levels of p53, PUMA, BAX, and p21 in OC cells after treatment with GV. GV treatment significantly increased the amount of p53, PUMA, BAX, and p21 proteins (Figure 5A,B). These data suggested that GV leads to apoptosis in OC cells by increasing cellular ROS levels and inducing the number of downstream molecules.

## 4. Discussion

The findings of this study represent a potential step toward repositioning of GV as a drug for the treatment of OC. Drug repositioning is the development of new indications for drugs already approved for the treatment of a disease. Drug repositioning has the advantage of being faster, cheaper, and less risky than the conventional drug development approach. These advantages are because drug safety and toxicity tests can be conducted more quickly in preclinical tests through the redevelopment of drugs that have already been developed and approved, and whose side effects in the human body are known through existing clinical trials. Sildenafil, sold under the brand name Viagra, is the best-known example of drug repositioning, which was originally developed as an anti-anginal drug, but now used for the treatment of erectile dysfunction. In particular, thalidomide is an antitumor drug developed through representative drug repositioning. Thalidomide was originally developed and used as an anti-emetic agent during pregnancy but was subsequently banned due to its side effects. It is now used as a multiple myeloma treatment through drug repositioning. Therefore, new drug development through drug repositioning is attracting attention as an alternative development strategy in the drug development process with a high probability of failure [20,21].

GV was first synthesized in 1861 and was widely used as an antibiotic until the discovery of penicillin in 1940. It is now mainly used for staining such as Gram staining. GV is known to have two modes of action. It is known to exhibit anti-angiogenic and antitumor effects in mammalian cells by inhibiting NADPH (nicotinamide adenine dinucleotide phosphate) oxidase, which converts oxygen into ROS. The other mode of action is induction of microbial cell death via formation of a covalent adduct with thioredoxin reductase 2. Currently, based on these effects, studies on the treatment of skin diseases caused by microbial infections and studies on its anti-angiogenic and antitumor effects are being reported [13,14]. Several studies have demonstrated the anticancer effect of GV. Specifically, treatment of breast cancer cells (MDA-MB-231) with GV was found to inhibit cell proliferation [19]; in addition, GV was shown to inhibit the proliferation of liver cancer cells (SK-HEP-1 and SMMC-7721) and cutaneous T-cell lymphoma cells (MyLa and SeAx) through the induction of apoptosis [15,18]. However, to the best of our knowledge, this is the first study to investigate the effect of GV on OC.

In this study, we performed TUNEL assay, and investigated the cleavage of PARP and caspase-3, which are important molecules for apoptosis, to verify whether inhibition of OC cell proliferation by GV was due to apoptosis. GV inhibited the proliferation of OVCAR8, SKOV3, A2780 cells in a dose-dependent manner (Figure 1). OVCAR8 and SKOV3 are cell lines of serous OC, and A2780 is a cell line of endometrial OC [22,23]. We observed that GV induced DNA fragmentation, caspase-3 activation, and PARP cleavage, known as an early apoptosis marker, in OVCAR8 and SKOV3 cells (Figure 3). In the intrinsic apoptosis pathway, cytochrome C is released from mitochondria by pro-apoptotic molecules and caspase-3 is activated through the cleavage of pro-caspase-3. Activated caspase-3 cleaves the PARP involved in DNA repair, thus inhibiting DNA repair during apoptosis (Figure 6). Moreover, GV increased TUNEL fluorescence intensity in OC cells, indicating increased DNA fragmentation (Figure 2). DNA fragmentation is a known marker of late apoptosis and is caused by activated caspase-3 [24]. Collectively, our results suggest that GV inhibits the proliferation of OC cells through the induction of apoptosis.

The chemical structure of GV is similar to that of diphenylene iodonium (DPI), a known NADPH oxidase (NOX) inhibitor [25]. NOX is a transmembrane protein that produces ROS and is known to be overexpressed in malignant tumors. In addition, ROS generated by NOX is known to be involved in cancer cell division, proliferation, and angiogenesis [26,27]. Xia et al. reported that DPI inhibits ROS production in OC cells, OVCAR3 [28].Unlike the previous study, we found that GV induced intracellular ROS levels in both OVCAR8 and SKOV3 cells (Figure 4). Several studies have shown that GV increases ROS levels in liver cancer cells (SK-HEP-1 and SMMC-7721), cutaneous T-cell lymphoma cells (MyLa and SeAx), and melanoma cells (A375) [15,17,19]. GV also appears to induce apoptosis and inhibit growth by increasing ROS in these cancer cells. p53 is known to be involved in numerous cellular activities such as cell cycle arrest, apoptosis, senescence, DNA repair, and cell differentiation. p53 is called a tumor suppressor protein because it induces apoptosis upon DNA damage and oncogene activation. However, in approximately 50% of human cancers, p53 is mutated, resulting in reduced expression, loss of function, or dysfunction [29,30]. In addition, it is known that the function of p53 as a transcription factor is inhibited by NOX in cancer cells [31]. Chen et al. and Garufi et al. reported that GV inhibits NOX and upregulates p53 in liver cancer cells (SK-HEP-1 and SMMC-7721), lung cancer cells (H1299), and colon cancer cells (RKO and HCT116) [15,16]. It has also been reported that PUMA is mainly located in the mitochondria, induces apoptosis by ROS production and increases erythroid 2-related factor 2 (NRF2) and heme oxygenase-1 (HO-1) levels in transfected A2780 and SKOV3 cells. The increased production of ROS activates NRF2 signaling to induce the expression of antioxidant enzymes, including HO-1, catalase, glutathione peroxidase, and superoxide dismutase, protecting cells against oxidative stress. Thus, PUMA induces ROS generation, leading to DNA damage and ultimately to cell death, although it may also enhance NRF2/HO-1 expression to protect against ROS-mediated oxidative stress [3]. In this study, GV upregulated p53, and also increased PUMA, BAX, and p21 regulated by p53 in OC cells (Figure 5). Collectively, our data suggest that GV induces cellular stress by increasing ROS and induces apoptosis by upregulated p53 in OC cells. However, the precise mechanisms of GV-induced ROS increase and its relationship with p53 in OC cells require further investigation.

In conclusion, this study explored the potential expansion of the indication of GV through drug repositioning. GV inhibited cell proliferation through apoptosis and significantly increased ROS and p53 in OC cells. This is the first study to demonstrate the effects of GV on OC cells and suggests the potential of GV as a novel therapeutic agent for OC.

## Figures and Tables

**Figure 1 biomedicines-11-01657-f001:**
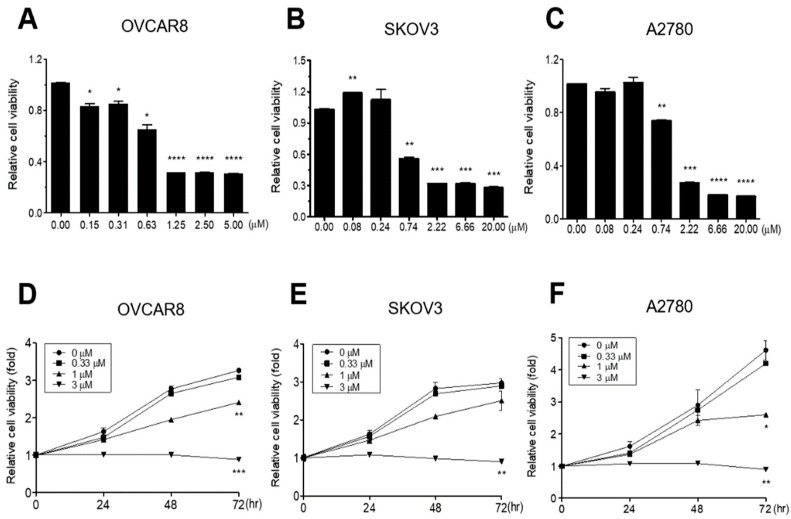
Gentian violet (GV) inhibited the proliferation of ovarian cancer (OC) cells. (**A**) OVCAR8, (**B**) SKOV3, or (**C**) A2780 cells were seeded onto 96-well tissue culture plates and treated with the indicated concentrations of GV for 72 h. (**D**) OVCAR8, (**E**) SKOV3, or (**F**) A2780 cells were treated with 0, 0.33, 1, or 3 μm of GV for 0, 24, 48, or 72 h. Relative cell proliferation rates were determined by WST-1 assay monitoring the mitochondrial succinate reductase activity at 450 nm. Data are shown as mean ± SEM duplicates from one of two independent experiments. The p-value was calculated by Student *t*-test (* *p* < 0.05, ** *p* < 0.01, *** *p* < 0.001, and **** *p* < 0.0001).

**Figure 2 biomedicines-11-01657-f002:**
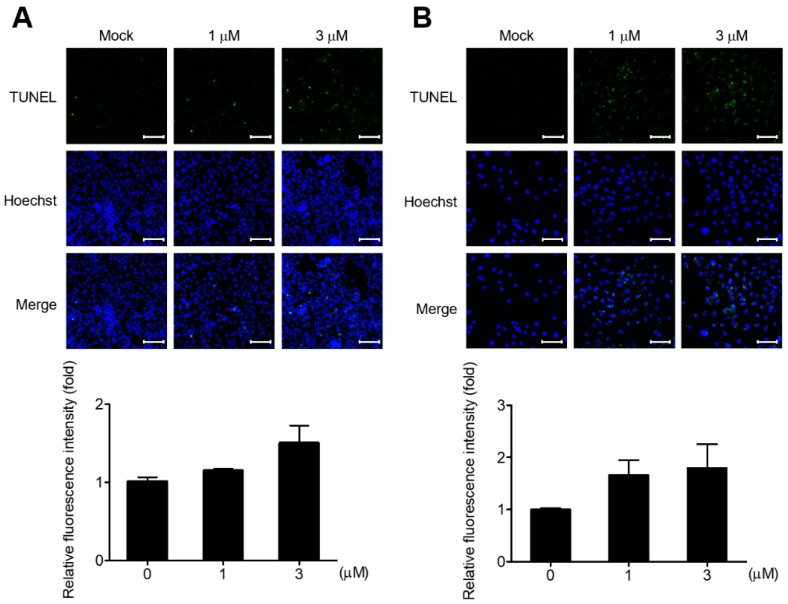
GV-induced apoptosis in OC cells. Apoptotic cells were examined using the TUNEL assay for DNA fragmentation. (**A**) OVCAR8 and (**B**) SKOV3 were treated with 0, 1, or 3 μm of GV for 48 h and imaged by confocal microscopy. TUNEL-positive nuclei are shown in green and total nuclei stained with Hoechst 33342 are shown in blue (upper). Relative TUNEL intensities versus negative control were quantified in three randomly selected microscopic fields (lower). Data are shown as mean ± SEM from one of two independent experiments. Scale bar = 100 μm.

**Figure 3 biomedicines-11-01657-f003:**
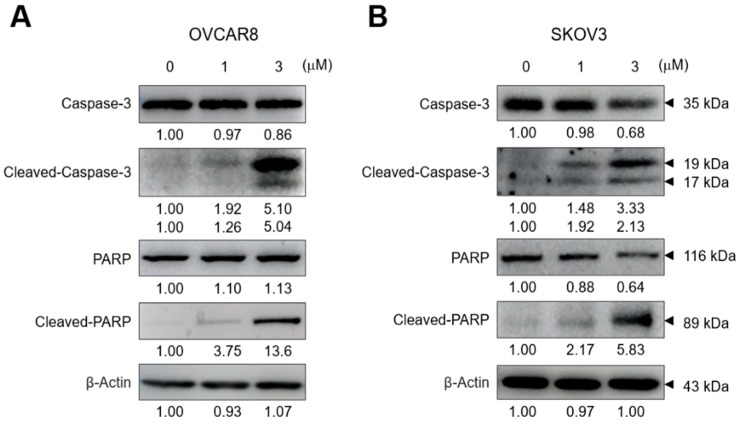
GV induced the cleavage of PARP and caspase-3 in OC cells. (**A**) OVCAR8 and (**B**) SKOV3 were treated with 0, 1, or 3 μm of GV for 24 h and whole-cell lysates were immunoblotted with antibodies against caspase-3, cleaved-caspase-3, PARP, cleaved-PARP, or β-actin. Results are representative of at least three independent experiments.

**Figure 4 biomedicines-11-01657-f004:**
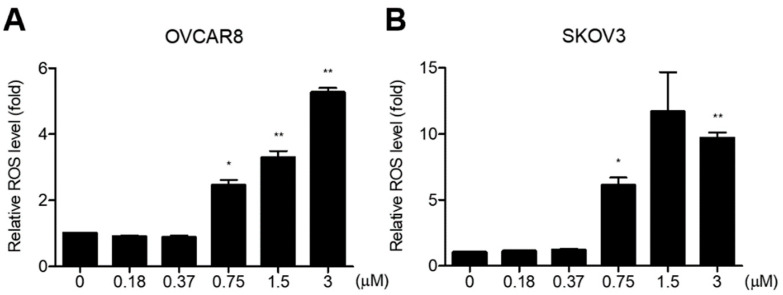
GV increased ROS level in OC cells. (**A**) OVCAR8 and (**B**) SKOV3 cells were treated with the indicated concentrations of GV for 24 h and stained with 10 μm H_2_DCFDA. ROS levels were measured with a microplate reader at 485/530 nm. Data are shown as mean ± SEM duplicates from one of two independent experiments. The p-value was calculated by Student’s *t*-test (* *p* < 0.05 and ** *p* < 0.01).

**Figure 5 biomedicines-11-01657-f005:**
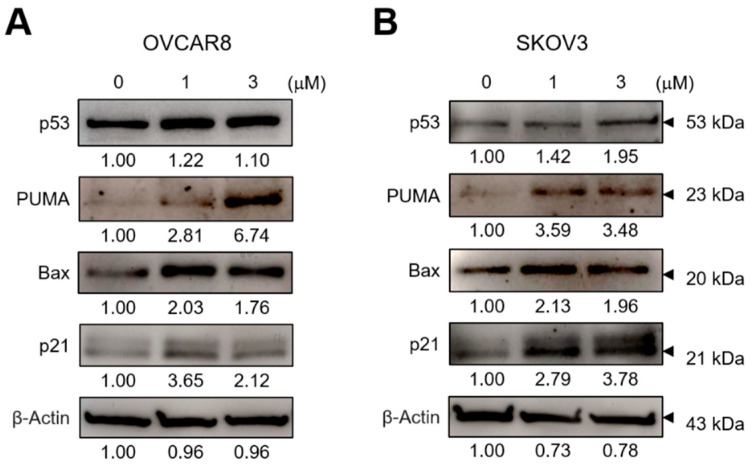
GV upregulated p53, PUMA, BAX, and p21 in OC cells. (**A**) OVCAR8 and (**B**) SKOV3 were treated with 0, 1, or 3 μm of GV for 24 h and whole-cell lysates were immunoblotted with antibodies against p53, PUMA, BAX, p21, or β-actin. Results are representative of at least three independent experiments.

**Figure 6 biomedicines-11-01657-f006:**
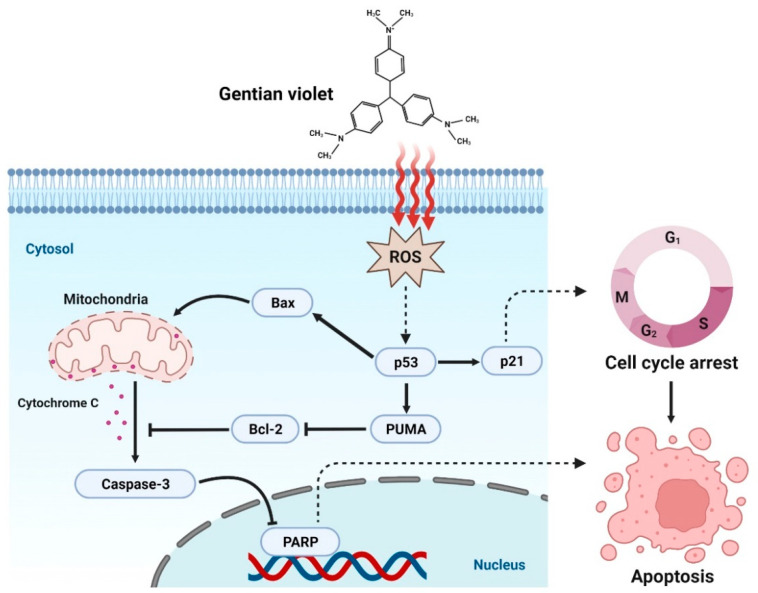
Schematic illustration of the mechanism by which GV induces apoptosis in OC cells. GV increases ROS and upregulates p53, which increases PUMA, BAX, and p21. Increased PUMA and BAX release cytochrome c from mitochondria and consequently induce cleavage of caspase-3 and PARP, inducing apoptosis.

## Data Availability

The data presented in this study are available on request from the corresponding authors.

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
