# Peer review of "Gentian Violet Inhibits Cell Proliferation through Induction of Apoptosis in Ovarian Cancer Cells"

_biomedicines, 2023, doi:10.3390/biomedicines11061657_

Round 1

Reviewer 1 Report

In this work, Authors have investigated the anticancer efficacy of Gentian Violet (GV) in ovarian cancer cell lines. They found that cell growth inhibition of the three cell lines by GV correlated to the induction of apoptosis, increased ROS and upregulated the expression of p53, PUMA, BAX, and p21.

Thus, Authors conclude that GV behaved a novel anti-proliferative agent, deserving exploitation in therapeutic strategies against ovarian cancer.

The work appears well conducted, and the results obtained with proper methods, support the conclusion and justify the rationale.

However, some points need to be more explained.

1. What is the veicol (solvent) for GV? How were the GV solutions obtained? Specify.

2. In addition, were the solutions obtained with the different concentrations of GV colored? If yes, please specify if this interfered with the spectrophotometric WST-1 assay at 450 nm.

3. A2780 cells have been used only for data depicted in figure 1, then only the other two cell lines. Why?? Is there a particular reason related to these cells?

4.  the antitumor activity appears to be well proven on these tumor cells. But what is the activity in non-proliferative/non-tumorigenic cells? That is, what is the selectivity of GV?

Reviewer 2 Report

In this manuscript, the authors reported the outcomes and conclusion of their study in which they determine the mechanisms by which Gentian (GV) inhibits proliferation and growth in ovarian cancer cell lines. In their studies they showed that GV inhibits proliferation in three ovarian cancer cell lines. They further showed that GV induces apoptosis via activation of caspase 3, and induces ROS production and increases expression of p53, Bax, PUMA and other pro-apoptotic proteins. Based on their results they provided a model which explains the mechanisms by which GV induces inhibition of ovarian cancer proliferation. Overall, their experimental methodologies are standard and acceptable. Their data presentation and interpretation are particularly good. Their conclusion is based on their experimental results. The manuscript is written nicely and not difficult to follow and understand. Overall, their manuscript will add new information to the field and will attract readers. However, there some important questions for the authors to answer and addressing those questions will likely improve the overall quality of the manuscript.

Concerns:

1.       You did not indicate the number of times you performed the experiments whose results you showed in all figures 1 to 5. Please, indicate.

2.       There are no p-values for your results shown in figures 1, 2, and 3. Did you determine the p-values? If so, provide them with those results.

3.       In your experiments whose results you showed in Figure 1, you included results from A2780 cell line. However, you did not show results from this cell line in figures 2, 3, 4, and 5. But you did not explain your rationale for not including the A2780 cells in the subsequent experiments. Please, provide the rationale.

4.       You showed that GV induces increases in ROS production, and promotes expression of p53, Bax, and other pro-apoptotic proteins. Based on the results you indicated in your model in figure 6 that p53 is downstream ROS production in the ovarian cancer cell lines. Do you have any direct evidence to support this conclusion? For example, did you pre-include some of the cells with compounds that block or inhibit ROS production to see whether blocking ROS production results in inability of GV to induce p53 expression?

5.       Did you examine the effects of GV on early-stage caspases such as caspase2 and caspase8, which could play a role in caspase3 activation?

Minor editing will be needed,

Reviewer 3 Report

The study is interesting and generally well written. Only minor changes are needed. In particular: 

Introduction: A short introduction on PUMA protein should be added since it has been involved in ROS and NRF2 pathway activation (see PMID: 35453348) ( I personally suggest to investigate NRF2/KEAP1 signalling in Gentian violet treated cells). 

Lines 28-39: It deserves to be pointed out that the high mortality rate of ovarian cancer is also due to chemoresistance occurrence (see PMID: 35453348).

4.4. Western blot analysis: Authors must report the product codes and dilutions of the primary antibodies used

Figures: Statistical differences must be shown with asterisks

Figure 3 and 5: Destitometric analysis must be reported. Molecular weights must be added

An accurate revision of punctuation is recommended

Round 2

Reviewer 1 Report

In the revised version, Authors have satisfactorily replied to the questions raised.